# Tracking of Dietary Patterns in the Secondary Prevention of Cardiovascular Disease after a Nutritional Intervention Program—A Randomized Clinical Trial

**DOI:** 10.3390/nu14224716

**Published:** 2022-11-08

**Authors:** Amanda Gonçalves Lopes Coura, Adélia da Costa Pereira de Arruda Neta, Rafaela Lira Formiga Cavalcanti de Lima, Ângela Cristine Bersch-Ferreira, Bernardete Weber, Rodrigo Pinheiro de Toledo Vianna

**Affiliations:** 1Postgraduate Program in Nutrition Sciences, Federal University of Paraíba, João Pessoa 58059-900, Brazil; 2Research Institute-HCor, São Paulo 04004-030, Brazil

**Keywords:** dietary patterns, healthy dietary pattern, unhealthy dietary pattern, cardiovascular disease

## Abstract

Individuals with a history of previous cardiovascular events have an increased risk of mortality and morbidity, so adherence to a healthy dietary pattern is essential. We aimed to evaluate and compare dietary patterns between the control and the experimental group from the BALANCE Program. A total of 2360 individuals aged 45 years or older with previous cardiovascular disease were included. The individuals were randomized into two groups: intervention (dietary prescription with nutritional recommendations, nutritional education program based on playful strategies, suggestions of typical and accessible Brazilian foods and intensive monitoring) and control (conventional nutritional counseling). The dietary patterns were identified using factor analysis with the principal component extraction method, and the t-Student tests and ANOVA test were performed to evaluate the associated factors. Four dietary patterns were identified for both groups: “Traditional”, “Snack”, “Western”, “Cardioprotective”. There was an increase in the variances of the “Cardioprotective” pattern in both groups. Regarding the “Western” pattern, there was a significant reduction in the variances of the experimental group (10.63% vs. 8.14%). Both groups had improvements in eating habits, especially in the first year of follow-up. The greater increase in adherence to the traditional and cardioprotective pattern in the experimental group justifies the initiative of the BALANCE program.

## 1. Introduction

Cardiovascular diseases (CVD) represent one of the greatest public health concerns because they are related to high premature morbidity and mortality and an overload of health systems, generating great socioeconomic costs worldwide [1,2,3]. According to the World Health Organization (WHO), CVD is the leading cause of death and accounted for 16% of total deaths from all causes in the world in 2019 [4]. In Brazil, CVDs were responsible for 30% of deaths [5]. A study shows a trend towards an increase in deaths from CVD and stroke in Brazil, which are the main causes of death in the country [6]. 

Dietary habits influence cardiovascular risk through an effect, for example, on blood pressure (BP), blood lipids, obesity, inflammation and endothelial function [7]. The likelihood of developing CVD is associated with eating patterns that are considered unhealthy (i.e., excessive intake of sodium and processed foods; added sugars; unhealthy fats; low intake of fruits and vegetables, whole grains, fiber, legumes, fish and nuts), along with a lack of exercise, overweightness and obesity, stress, alcohol consumption or smoking [8]. A healthy diet is the cornerstone of cardiovascular disease (CVD) prevention.

Individuals with a history of previous cardiovascular events are more likely to develop new events and have an increased risk of mortality, with secondary prevention being decisive to increase survival. Thus, it is extremely important to implement lifestyle changes on this population, such as adhering to a diet composed of foods that are rich in fiber and antioxidant nutrients, is associated with a reduction in the consumption of saturated fats and ultra-processed foods and the adoption of healthy lifestyle habits, considering personal preferences, in addition to cultural and economic aspects [1,2,3,9,10,11,12,13,14,15].

However, nutritional epidemiology studies that analyze the relationship between diet and disease based on the consumption of isolated foods or nutrients are limited, since, in practice, individuals consume combinations of various foods and nutrients that can interact with each other, which makes the human diet a complex and dynamic exposure factor [15,16,17,18,19,20]. In this sense, the WHO recommends the analysis of dietary patterns, which assess a set or group of foods consumed by a given population [16,19,20].

Thus, the aim of the present study was to compare dietary patterns between the control group and the experimental group before and after the intervention of the BALANCE program.

## 2. Materials and Methods

### 2.1. Study Design and Population

This is a sub-study of the multicenter randomized clinical trial Brazilian Cardioprotective Food Program (, BALANCE), coordinated by the Hospital do Coração in partnership with the Basic Health System Development Support Program (PROADI-SUS) of the Ministry of Health. Thirty-five reference hospitals in cardiology in Brazil participated in the study [21,22].

In the period between March 2013 and April 2015, 2534 participants were randomly selected and followed until October 2017. Clinical trial registration: NCT01620398.

The inclusion criteria in the BALANCE study were individuals aged 45 years or older with one or more of the following indicators of established CVD: (a) coronary artery disease (defined by previous acute myocardial infarction; stable or unstable angina; history of atherosclerotic stenosis ≥70% of the diameter of any coronary artery on conventional or computed tomography coronary angiography; or history of angioplasty, stenting or coronary artery bypass surgery); (b) previous stroke; and (c) peripheral vascular disease (ankle-to-arm ratio <0.9 of systolic blood pressure in either leg at rest, angiography or Doppler demonstrating >70% stenosis in a noncardiac artery, intermittent claudication, vascular surgery for atherosclerotic disease, amputation due to atherosclerotic disease or aortic aneurysm) [22].

The exclusion criteria were as follows: neurocognitive or psychiatric conditions that may hinder the collection of reliable clinical data (defined at the investigator’s discretion), life expectancy of less than 6 months (e.g., metastatic malignancy or other factor defined at the investigators’ discretion), pregnancy or lactation, liver failure with a history of encephalopathy or anasarca, renal failure with indication for dialysis, congestive heart failure, previous organ transplantation, wheelchair use or any restrictions to receiving an oral diet [22].

The population considered for this sub-study was composed of study participants at the initial time (baseline year), after one year and after two years of the intervention of the BALANCE program.

The sample planning and protocol of the study were previously published [22]. The participants were randomly distributed (in a 1-to-1 ratio) into two groups: the experimental group and the control group. The randomization was performed in blocks with stratification by study site. Allocation concealment was guaranteed through a central web-based automated system. Outcome adjudicators and statisticians were blinded to the assigned interventions. The protocol study describes the randomization and blinding process [22].

### 2.2. Intervention

#### 2.2.1. Control Group

Participants in the control group were directed to follow the conventional nutritional advice developed by nutritionists and based on a diet that is low in fat, energy density, sodium and cholesterol, in accordance with the guidelines for the treatment of CVD [12,22]. Dietary prescriptions were qualitative, and participants were given a list of foods that should be preferred or avoided [21,22].

#### 2.2.2. Experimental Group

The details of the intervention in this group were reported in the study protocol [21]. The participants of this group received a food plan prescription according to the BALANCE Program, which contemplated nutritional guidelines based on Brazilian and international guidelines for dyslipidemias and cardiovascular diseases, guided by the nutritional composition of the foods, such as the Mediterranean diet and the DASH diet, but with the proposals for the adequate intake of nutrients considering those foods that are consumed and accessible in Brazil [21,22].

The BALANCE program was based on three concepts: dietary prescription with nutritional recommendations according to Brazilian guidelines for the treatment of cardiovascular diseases; a nutritional education program based on playful strategies and with suggestions of accessible and typical Brazilian foods; and an intensive follow-up through face-to-face visits, group sessions and telephone calls [21,22].

For the dietary prescription, foods were categorized according to energy and nutrient density (saturated fatty acids, cholesterol and sodium). Foods with energy density ≤ 1.11 kcal/g, a saturated fatty acid density ≤ 0.01 g/g, a cholesterol density ≤ 0.04 mg/g and a sodium density ≤ 2.01 mg/g were included in the “green” group. Foods with one or two nutrient densities above the cutoff points were included in the “yellow” group and those with three or four nutrient densities above the cutoff points were included in the “blue” group. The “red” group was composed of foods with trans fatty acids, artificial sweeteners and preservatives [21,22].

The colors mentioned above were the foundation for the participants’ food guide proposed by the BALANCE Program, which used the Brazilian flag as a reference: green occupies the largest area of the flag, indicating that the foods of the green group should be more strongly represented in the diet. Yellow is the second-most abundant color of the flag, suggesting that this food group should be less prevalent in the diet. Blue, on the other hand, covers only a small part of the flag, which suggests the restricted intake of this food group. Finally, the absence of the red color on the flag reinforces the recommendation that foods belonging to this group should not be consumed [21,22].

### 2.3. Dietary Assessment

Food consumption data were obtained through the 24 h recall (R24h) in each biannual face-to-face meeting conducted by a previously trained researcher. The participant reported all the food and drinks they had consumed in the last 24 h. To facilitate the quantification of food consumption, the interviewers relied on a photo album containing homemade measurements and portions. Two R24hs were applied before the intervention, and one R24h was applied at each biannual meeting [21,22], totaling six food consumption assessments throughout the follow-up for each participant. The data obtained were recorded in the Nutriquanti software [23], which prioritizes Brazilian nutritional composition tables.

For the longitudinal analysis of the dietary pattern, two R24hs were associated in each analysis moment to ensure the reliability of the responses and reduce intrapersonal variation: in the baseline year (two R24hs applied before the start of the nutritional intervention), in the first year of follow-up (the 6-month and 12-month follow-up meetings) and in the second year of follow-up (the 18-month and 24-month follow-up meetings). The association of the R24hs met the alternating days criterion, enabling the estimation of the usual intake of the population [24,25].

### 2.4. Dietary Patterns

The food and beverages reported in each R24h were allocated into groups based on the consumption characteristics and food nutritional composition.

After this step, the usual intakes of the individuals of the experimental group and the control group were obtained through the Multiple Source Method (MSM), a web tool developed by researchers of the European Prospective Investigation on Cancer and Nutrition that estimates the usual intake of food consumed by populations and individuals [26].

The MSM method is performed in three stages and requires at least two days of dietary measurements with R24hs in a subsample of the target population, which, in the case of this study, was 52% of the individuals. MSM uses logistic regression to estimate the food intake probability for each individual and then uses linear regression to estimate the usual amount of food intake in the days of consumption. After this process, the method calculates the usual food intake by multiplying the results of the first step with those obtained in the second step [27].

The dietary patterns were identified using factor analysis with the principal components extraction method [28]. The suitability of the data for factor analysis was verified through the Kaiser–Meyer–Olklin test (KMO) and the Barlett sphericity test [28]. Eigenvalues greater than 1.0, the eigenvalues graph (Screp test) and the interpretability of the patterns [28] were used to identify the number of patterns to be retained. The varimax orthogonal rotation was performed to generate uncorrelated factors, facilitating the interpretation of the findings. The highest factor load values were considered to name the patterns. Subsequently, the scores of each pattern were obtained, representing the adherence of the individuals to each identified pattern.

The procedures were performed for each time (baseline year, first year of follow-up and second year of follow-up) and for each group (experimental and control) in order to see the changes in the patterns throughout the follow-up in each group. Factor loads ≥ 0.30 or ≤ 0.30 were considered as the cut-off points.

### 2.5. Covariate Assessment

Data related to socioeconomic, anthropometric, biochemical and lifestyle factors, the presence of comorbidities, evidence of manifest atherosclerosis and food consumption were collected through face-to-face consultations with previously trained professionals, according to all the procedures defined in the study [21,22].

Socioeconomic data, such as those relating to gender (male or female), age (45 to 59 years or older than 60 years) [29], socioeconomic classification (Class A and B or Class C, D and E) [30], education level (high—high school or higher—or low—illiterate/ incomplete elementary school, complete elementary school) and region of the country (North and Northeast or Midwest, South and Southeast), were collected at the initial time of the study [21,22].

Anthropometric data were collected at each biannual meeting: weight and height were used to estimate the body mass index (BMI-weight/(height)2), which was classified as a risk when the BMI was greater than or equal to 25.0 kg/m^2^ for adults [31] and when the BMI was greater than or equal to 28 kg/m^2^ for the elderly [32]. Waist circumference (WC) was classified as a risk when it was greater than or equal to 94 cm for men and greater than or equal to 80 cm for women. All measurements were performed in duplicate, and the mean of the two measurements was considered [21,22,33].

Biochemical tests were also requested in each biannual meeting, with the following categorizations: total cholesterol (greater than or equal to 190 mg/dL or less), LDL-cholesterol (greater than or equal to 130 mg/dL or less), HDL-cholesterol (less than or equal to 40 mg/dL or less), triglycerides (greater than or equal to 150 mg/dL or less) and fasting glucose (greater than or equal to 100 mg/dL or less) [21,22].

The collected lifestyle variables were smoking (yes or no) semiannually and sedentary lifestyle [34,35] (yes or no) annually.

The diagnosis of comorbidities, such as systemic arterial hypertension (SAH) (yes or no), diabetes mellitus (DM) (yes or no) or dyslipidemias (DLP) (yes or no), was obtained from the medical records or reported by the patient’s doctor [21,22].

Individuals were classified with respect to the presence or absence of the following conditions: asymptomatic CAD, symptomatic CAD (history of angina), treated CAD (previous angioplasty), AMI, stroke, asymptomatic PAD, symptomatic PAD (intermittent claudication), treated PAD (vascular surgery for atherosclerotic disease) and amputation due to arterial or aortic aneurysm [21,22].

### 2.6. Statistical Analyses

The comparative analyses were made based on the repeated panels model, where each food pattern of each group at each time was estimated considering the total of available and complete participants at the respective times in order to increase the consistency of the patterns obtained.

To evaluate the factors associated with the dietary patterns, the t-Student test and analysis of variance were performed, comparing the means of the scores for each group, divided according to the factors of interest. A significance of 5% was considered for all analyses.

All statistical analyses were performed with the Stata SE software, version 13 (StataCorp LLC, College Station, TX, USA).

### 2.7. Ethical Aspects

All participants read and signed the Informed Consent Form prior to their inclusion in the study. The study is registered at www.clinicaltrials.gov (accessed on 10 August 2022) under the protocol number NCT01620398. The study was approved by the Ethic Committee of the University Hospital of the Federal University of Campina Grande on 28 August 2012 (Number 84275).

## 3. Results

At the initial time, called the baseline year, 2357 individuals participated. A total of 175 participants (6.9% of the total sample recruited) were not included due to incomplete data on food consumption. In the first year of follow-up, data from 1807 individuals from both groups remained, and in the second year of follow-up, data from 1354 individuals from both groups remained (Figure 1).

Regarding the cardiovascular pathologies presented by individuals undergoing secondary prevention included in the study, 93% of individuals in the experimental group and 92% of individuals in the control group had coronary artery disease; 12% in each of the two groups had a previous acute myocardial infarction; and 12% in each of the two groups had previous peripheral vascular disease [22].

In the baseline year, before the intervention, 52% of the individuals answered a second R24h. In the first year of follow-up, 90.6% of the individuals answered two R24hs, and in the second year of follow-up, 63% of the individuals answered two R24hs. The group of participants with complete recalls was compared with the other participants who failed to perform a recall to see if there was any bias in these information losses.

Table 1 shows the comparison of the sociodemographic and health characteristics of the individuals according to the study’s participation groups in the baseline year and second year of follow-up, revealing that the groups were similar in the first evaluation and remained so as the study progressed. The comparison of each group at different times of the study was also carried out and showed that there were no significant differences, revealing that the follow-up losses were random (data not presented).

Of the total number of participants, 58.4% were men and 57.7% were aged between 45 and 65 years. In relation to the economic class, 71.1% belonged to classes A or B; as for comorbidities, 90.1% of the participants had SAH and 44.3% had DM. Regarding nutritional status, 73.1% were overweight and 95.2% had waist circumference as a risk for CVD (Table 1).

The groups formed after classifying the foods and beverages consumed, according to their consumption characteristics and nutritional composition, are shown in Table 2. Fifteen large groups were formed, as described in the table.

The factor loads for each food pattern identified in each group and the proportion of the variation explained by these patterns at each time are presented in Table 3 (experimental group) and Table 4 (control group).

Table 3 shows that four main dietary patterns were identified at the baseline year for the **experimental group** before they received the nutritional counseling and prescription program. The first food pattern was called “*Traditional*”, which had high factor loads for rice, beans, meats and eggs, in addition to a negative load for pasta. The second pattern was characterized by positive loads for pasta, processed meats, sugary drinks and sugary juices, snacks and sweets and desserts, in addition to high negative loads for fruits and natural juices without sugar. This pattern was called “*Western*”. The third pattern, called “*Snack*”, was composed of positive loads for breads, cereals and crackers, dairy products, butter and margarine. The fourth pattern was called “*Cardioprotective*”. It was characterized by positive loads for vegetables, fruits and natural juices without the addition of sugar, olive oil and cheeses. The four patterns explained 38% of the population’s consumption variance.

In the first year of follow-up, the same four dietary patterns were identified in the **experimental group**; however, there were changes in the factorial loads of the foods and the proportion of participants in each consumption quartile, characterizing changes in the eating habits of the individuals. The “*Cardioprotective*” food pattern, with positive loads for vegetables, fruits and natural juices without the addition of sugar and olive oil, became the second pattern, and the pattern “*Western*”, which is considered unhealthy, became the third pattern (Table 3).

In the second year of follow-up, there were new changes in the factorial loads of the foods and in the proportion of participants in the **experimental group** in each consumption quartile. The “*Cardioprotective*” food pattern, with positive loads for vegetables, fruits and natural juices without the addition of sugar and olive oil, became the third pattern, while the “*Western*” pattern became the fourth pattern (Table 3), demonstrating that the nutritional intervention over the two years promoted a continuous reduction in the consumption of ultra-processed foods and sugary drinks, with an increased intake of vegetables, fruits and olive oil, representing a variance of 10.63%, 9.27% and, finally, 8.14% at the end of the follow-up.

With respect to the **control group**, similar dietary patterns to those identified in the experimental group were seen in the baseline year. The first food pattern identified in the **control group** was the “*Traditional*” one, which had high factor loads for rice, beans, meats and eggs. The second pattern was characterized by positive loads for breads, cereals and crackers, dairy products and butter and margarine and by negative loads for coffee and tea, and it was called “*Snack*”.

The third pattern was called “*Cardioprotective*”. It is characterized by positive loads for vegetables, fruits and natural juices without the addition of sugar, olive oil and cheeses. The fourth pattern was called “*Western*”, consisting of pasta, processed meats, butters and margarines, sugary drinks and juices with sugar, snacks and sweets and desserts. It also had a high negative load for fruits and natural juices without sugar. The four patterns explained 35% of the population’s consumption variance (Table 4).

In the first year of follow-up, the same four dietary patterns were identified. However, changes were observed in the factorial loads of the foods and in the proportion of participants in each consumption quartile, characterizing changes in the eating habits of participants in this group. The “*Cardioprotective*“ dietary pattern became the second pattern, while the “*Western*” pattern became the third (Table 4).

In the second year of follow-up, the dietary patterns “*Traditional*” and “*Cardioprotective*” remained as the first and second patterns, respectively, but there were changes in the factor loads of the other two food patterns. These changes revealed that the nutritional intervention in the control group also promoted important positive changes in the eating habits of the participants over the two years (Table 4).

Regarding the association of dietary patterns with the characteristics evaluated in the follow-up, a greater adherence to the “*Cardioprotective*” pattern was observed in the elderly male subjects of the experimental group during the baseline year and second year of follow-up when compared to women and adults (Table 5). In the second year of follow-up, there was a greater adherence to the “*Cardioprotective*” pattern among eutrophic individuals (according to the BMI analysis).

No statistical differences were observed in the adherence to the “cardioprotective” pattern among the participants of the experimental group regarding economic class, nutritional status, waist/height ratio, level of physical activity and smoking (Table 5).

Adult men participating in the control group showed greater adherence, during the baseline year, to the “*Cardioprotective*” pattern (Table 6). In the second year of follow-up, no statistical differences were observed in the adherence of the participants to the “*Cardioprotective*” pattern (Table 6). There were also no statistical differences in the adherence to this pattern with respect to economic class and nutritional status at baseline and after two years of follow-up (Table 6).

Since the cardioprotective pattern is recommended for the secondary prevention of cardiovascular disease and the western pattern is the least suitable for healthy eating, those characteristics that had the greatest relationship with these patterns at the beginning and at the end of the follow-up were observed in both groups (Table 5 and Table 6).

Initially, some characteristics were more related to the eating pattern, but after two years the impact of the intervention was independent, with no distinction between the characteristics under analysis. With respect to sex, this characteristic did not change over time, even though men adhered more to these two eating patterns (Table 5).

In the control group, the western pattern was initially more related to the characteristics of male sex, adulthood, physical activity and smoking. This pattern continued to be more adherent to these characteristics, except for physical activity. Initially, men had a greater adherence to the cardioprotective pattern, but after two years, they were no longer associated with any of the characteristics analyzed (Table 6).

## 4. Discussion

During the baseline year, the eating pattern that explained the eating habits of individuals in both groups the most was the pattern “*Traditional*”. This result corroborates the data found in the Family Budget Survey (*Pesquisa de Orçamentos Familiares*, POF 2017–2018), which evaluated the food consumption of a representative sample of the Brazilian population between 2017 and 2018 and revealed that the consumption frequencies of rice and beans (76.1% and 60.0%, respectively) were the highest among the foods reported by the participants of the survey and were the most frequently named items [36]. In addition, the phone survey of the Risk and Protection Factor Surveillance System for Chronic Diseases (*Vigilância de Fatores de Risco e Proteção para Doenças Crônicas por Inquérito Telefônico*, VIGITEL), which analyzed the food consumption of a representative sample of the Brazilian population by telephone, observed that 59.7% of the population reported consuming beans at least five times a week [37]. This is a cultural characteristic of the Brazilian population’s food intake.

The longitudinal analysis of the present study enabled the identification of the maintenance of this dietary pattern’s variance over the two years of follow-up. This is a positive result, since this pattern includes foods with an important amount of dietary fiber, a nutrient that is directly related to cardiovascular prevention [12,38,39,40].

The observed variation in the experimental group of the dietary pattern called “*Western*” revealed that there were benefits for the secondary prevention of CVD after the intervention of the Brazilian Cardioprotective Food Program, since the increased consumption of the foods identified in this pattern is associated with an increased risk of CVD [41,42,43,44,45].

Zhang et al. (2020) analyzed data from a representative sample of the population of the United States of America in order to evaluate the association between the intake of ultra-processed foods and cardiovascular health based on metrics of the American Heart Association, which has defined scores according to lifestyle information, BMI and biochemical tests. They observed an inverse association between ultra-processed food intake and cardiovascular health, with an adjusted OR of 2.57 for the highest consumption quartile [46].

In recent years, a high consumption of unhealthy foods has been observed in Brazil. The POF 2017–2018 observed that the consumption of ultra-processed foods (rich in saturated fats, trans fats and sugars and poor in fiber) accounted for 19.5% of the total calories consumed by adults and 15.1% of the calories among the elderly [36].

The VIGITEL study observed a frequency of 15% in terms of soda consumption on five or more days of the week [37]. This unfavorable scenario for the adoption of healthy eating practices makes food education policies and programs for the general population and the control of the advertising of unhealthy foods even more important, and all these measures become even more relevant when it comes to vulnerable groups, such as people in the secondary prevention of cardiovascular diseases.

The increase in the consumption of ultra-processed foods, which has been observed in the Brazilian population in recent years and is described in these population-based surveys, may explain the increase in the variance of the snack pattern, even in the experimental group, although this variance had an important reduction in the first year of the study’s follow-up.

Several studies have shown that dietary patterns that focus on foods that are considered healthy, such as cereals and whole grains, fruits, vegetables, oilseeds, fish and poultry, olive oil and vegetable oils and skimmed dairy products, and that restrict foods that are considered unhealthy, such as refined grains, added sugars, sugary drinks, red and processed meats and salt and trans fats, are associated with the primary and also secondary prevention of CVD and its comorbidities, and they could prevent up to three-quarters of the deaths caused by these diseases [47,48,49,50,51,52].

The pattern identified in this study called “*Cardioprotective*” was composed of such foods, and that is the reason for its name. At the initial moment, the variance observed for this pattern in the **experimental group** was 8.05%, with an increase of almost two percentage points after the first year. In the second year, there was a slight reduction of less than 0.5 percentage points, revealing that the program not only increased the consumption of vegetables, fruits and olive oils after the intervention but also maintained this behavior over time.

The **control group** had a variance of 9.04% for the same “*Cardioprotective*” standard at the initial moment, with an increase of a little over one percentage point in one year and a small reduction after two years of follow-up, showing that traditional nutritional orientation over time also proved important for maintaining healthy habits.

This study revealed, through a longitudinal analysis, that there was an improvement in the eating habits of the participants after the intervention, with an increase in the consumption variances of healthy foods.

The WHO recommends an intake of 400 g per day of fruits and vegetables in order to prevent various chronic diseases [9]. Data from the POF 2017–2018 showed that the intake of these foods by the Brazilian population fell short of the recommendation, in addition to reporting a reduction in their consumption in the last 10 years [39]. Only 22.9% of Brazilians meet WHO recommendations for fruit and vegetable intake, and 34.3% of the population regularly consumes (five or more days a week) these foods, according to data from the VIGITEL survey [37].

Understanding the changes in dietary patterns is important to base interventions with the aim of improving diet quality and good health outcomes, since this type of tracking can evaluate the stability or modification of these patterns over time [53,54].

Longitudinal tracking assesses the stability or maintenance of a variable over time and helps us understand longitudinal changes in the food consumption of individuals [53,54]. Studies aimed at tracking eating patterns are more common in children and adolescents [55,56], during the transition to adult life [57] and in obese individuals [54]. Little is known about changes in dietary patterns over time in individuals in the secondary prevention of CVD. This study is the first to investigate changes in dietary patterns over time among individuals in the secondary prevention of CVD.

The results of this study prove the importance of nutritional intervention with specialized professionals in promoting changes in eating and lifestyle habits, since the **experimental group** showed a higher increase in the variance of the “*Cardioprotector*” pattern than the control group after two years. Because this is a follow-up study, the maintenance of a healthy eating pattern could be observed, which can bring benefits for the prevention of new cardiovascular events, in addition to improving quality of life.

As a limitation, we highlight that it was not possible to identify the tracking coefficients of the dietary patterns because the changes in the total variances of the patterns identified in the period made it impossible to analyze these coefficients.

However, we highlight that this study analyzed data from a population in a very specific situation of secondary prevention of CVD, composed of adults and elderly people from five regions in Brazil who received nutritional intervention. Such characteristics are still scarce in the literature.

## 5. Conclusions

The results showed that both groups had improvements in eating habits, especially in the first year of follow-up. The greater increase in adherence to the traditional and cardioprotective patterns in the experimental group justifies the initiative of the BALANCE program. Knowledge of consumption patterns makes it possible to adjust the necessary interventions in the long term; however, only a longer follow-up can confirm the protective effect of these interventions.

## Figures and Tables

**Figure 1 nutrients-14-04716-f001:**
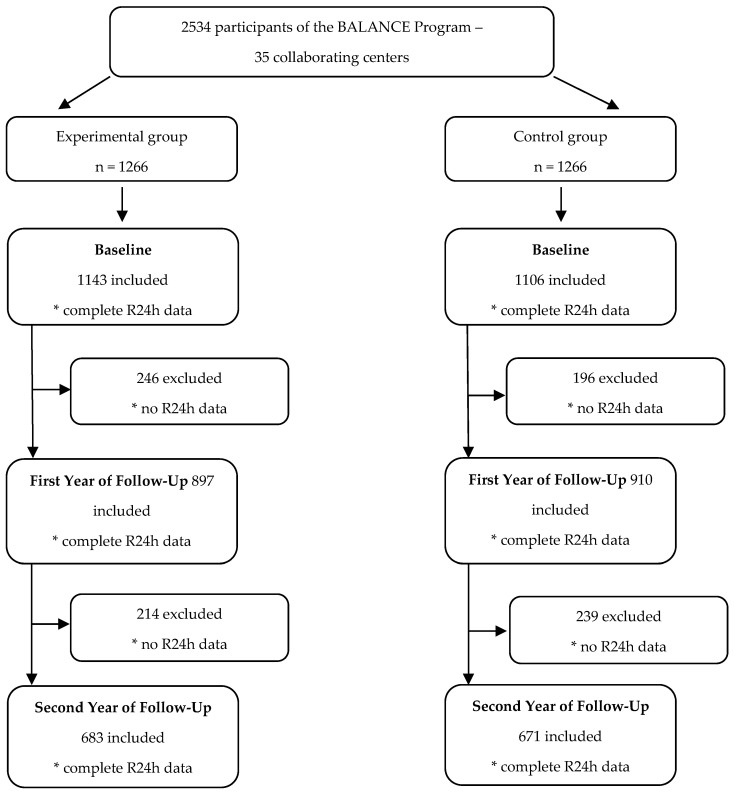
Eligibility, randomization and follow-up of the participants.

**Table 1 nutrients-14-04716-t001:** Sociodemographic and health characteristics of individuals in the secondary prevention for CVD at the initial time and after the second year of follow-up of the BALANCE program (n = 2360).

Variables	Baseline Year						After 2 Years of Follow-Up					
	Experimental		Control				Experimental		Control			
	n (%)		n (%)		*p*		n (%)		n (%)		*p*	
Sex												
Male	705 (51.2)		671 (48.8)		0.270 *		482 (51.1)		462 (48.9)		0.772 *	
Female	479 (48.9)		500 (51.1)				319 (50.3)		315 (49.7)			
Age												
Adult	684 (50.3)		676 (49.7)		0.983 *		456 (49.6)		464 (50.4)		0.247 *	
Elderly	501 (50.3)		496 (49.7)				346 (52.5)		310 (47.5)			
Economic Class												
A/B	717 (50.1)		714 (49.9)		0.840 *		475 (50.1)		473 (49.9)		0.832 *	
C/D/E	294 (50.6)		287 (49.4)				208 (50.7)		202 (49.3)			
Acute Myocardial Infarction												
No	587 (49.8)		592 (50.2)		0.635 *		410 (49.9)		411 (50.1)		0.478 *	
Yes	597 (50.8)		579 (49.2)				389 (51.7)		363 (48.2)			
Stroke												
No	1041 (50.3)		1030 (49.7)		0.978 *		702 (50.8)		679 (49.2)		0.935 *	
Yes	143 (50.3)		141 (49.7)				97 (50.5)		95 (49.5)			
Aortic Aneurysm												
No	1157 (50.2)		1149 (49.8)		0.495 *		782 (50.6)		763 (49.4)		0.289 *	
Yes	27 (55.1)		22 (44.9)				17 (60.7)		11 (39.3)			
Hypertension												
No	111 (47.8)		121 (52.2)		0.427 *		77 (45.3)		93 (54.7)		0.129 *	
Yes	1073 (50.6)		1048 (49.4)				722 (51.5)		681 (48.5)			
Diabetes Mellitus												
No	667 (50.9)		643 (49.1)		0.516 *		459 (51.6)		430 (48.4)		0.449 *	
Yes	517 (49.6)		526 (50.4)				340 (49.7)		344 (50.3)			
Total Cholesterol												
<190 mg/dL	803 (49.6)		817 (50.4)		0.160 *		455 (49.7)		461 (50.3)		0.635 *	
≥190 mg/dL	382 (52.8)		355 (47.2)				173 (51.2)		165 (48.8)			
LDL-Cholesterol												
<130 mg/dL	951 (50.1)		946 (49.9)		0.454 *		514 (49.8)		519 (50.2)		0.687 *	
≥130 mg/dL	234 (52.3)		226 (47.7)				93 (51.4)		88 (48.6)			
HDL-Cholesterol												
>40 mg/dL	576 (49.1)		598 (50.9)		0.155 *		349 (49.9)		350 (50.1)		0.851 *	
≤40 mg/dL	571 (52.0)		526 (48.0)				270 (50.5)		265 (49.5)			
Triglycerides												
<150 mg/dL	617 (49.0)		642 (51.0)		0.097 *		353 (49.5)		360 (50.5)		0.592 *	
≥150 mg/dL	533 (52.5)		482 (47.5)				266 (51.1)		255 (48.9)			
Glycemia												
<100 mg/dL	498 (52.0)		460 (48.0)		0.216 *		281 (51.6)		263 (48.4)		0.342 *	
≥100 mg/dL	648 (49.3)		665 (50.7)				349 (48.9)		364 (51.5)			
Body Mass Index												
Underweight	25 (41.7)		35 (58.3)				16 (53.3)		14 (46.7)		0.428 *	
Eutrophic	278 (48.5)		295 (51.5)		0.196 *		149 (47.1)		167 (52.9)			
Overweight/obesity	877 (51.3)		831 (48.7)				502 (51.2)		478 (48.8)			
Waist Circumference												
Adequate	53 (47.3)		59 (52.7)		0.521 *		32 (48.5)		34 (51.5)		0.753 *	
Inadequate	1122 (50.4)		1133 (49.6)				639 (50.5)		627 (49.5)			
Physical Activity												
Active	367 (49.5)		374 (50.5)		0.573 *		448 (48.2)		481 (51.8)		0.012	
Inactive	727 (50.8)		704 (49.6)				226 (55.7)		180 (44.3)			
Smoking												
Never smoked	461 (51.4)		436 (48.6)		0.638 *		326 (53.3)		286 (46.7)		0.198 *	
Former smoker	632 (*50.1*)		634 (*49.9*)				428 (*49.5*)		437 (*50.5*)			
Smoker	87 (*47.8*)		95 (*52.2*)				44 (*45.4*)		53 (*54.6*)			

* *p* values not significant according to the Chi-square test.

**Table 2 nutrients-14-04716-t002:** Grouping of foods and recipes for the analysis of the dietary pattern of individuals.

Food Groups	Description
Rice and recipes	Rice and rice-based recipes
Olive oil	Extra virgin olive oil
Sugary drinks	Concentrated and pasteurized juices, natural and pulp juices with sugar, powdered juice, ready-made teas, soft drinks, milk with chocolate, chocolate drink, chocolate, fruit vitamin with sugar, milk with sugar (with or without coffee)
Meat and eggs	Beef, pork, goat meat, poultry and their recipes, eggs
Processed meats	Sausages, cold meats and canned meat
Cereals, breads and crackers	Polenta, couscous, pamonha (corn-paste wrapped in husks), cooked corn, oats, granola, quinoa, flaxseed, cereal bar, morning cereals with sugar, sugary children’s cereals, corn starch, porridges, salted biscuits, breads, sweet bread, stuffed bread, toast, panetone, sweet and homemade cookies, cakes
Sweets	Added sugar, sweets, chocolates, desserts, homemade and industrialized sweets
Beans	Beans and legumes
Fruits, natural juices	Fruit, dried fruit, natural juices or unsweetened pulp
Snacks	Pizza, sandwiches, chips.
Dairy products	Unsweetened milks, unsweetened fruit vitamin, yoghurts, dairy drinks
Butter and margarine	Butter and margarines with and without salt
Pasta	Pasta with or without sauce, stuffed pasta
Cheeses	White, yellow and processed cheeses
Vegetables	Raw, cooked or fried vegetables and their recipes

**Table 3 nutrients-14-04716-t003:** Food patterns and factor loads of food groups extracted from the factor analysis with the principal component extraction method for individuals of the BALANCE program’s intervention group. Brazil.

Food Groups	Food Patterns—Baseline Year				Food Patterns—First Year of Follow-Up				Food Patterns—Second Year of Follow-Up			
	Traditional	Western	Snack	Cardioprotective	Traditional	Cardioprotective	Western	Snack	Traditional	Snack	Cardioprotective	Western
	Factor Load *											
Rice	0.8193	0.0002	−0.0289	0.0451	0.7367	0.0247	−0.0293	−0.0807	0.7833	0.0120	0.0937	−0.0058
Pasta	−0.3068	0.3377	0.0585	0.0265	−0.0960	−0.2391	0.3637	0.0703	−0.4666	−0.0135	0.2739	0.1628
Beans	0.6922	−0.0085	0.0596	−0.0004	0.6793	0.0572	−0.1083	0.0581	0.6356	0.1701	0.2042	0.0200
Vegetables	0.1771	−0.0806	−0.0556	0.6897	0.2769	0.6732	−0.1340	−0.0299	0.1846	−0.0154	0.5930	−0.0623
Fruits and juices	−0.2003	−0.5305	0.0277	0.3315	−0.2562	0.3863	−0.5068	0.0351	−0.1488	0.0036	0.2557	−0.4343
Meat and eggs	0.4875	0.2725	0.0157	0.2178	0.6072	0.1762	0.3021	−0.0317	0.3965	−0.1931	0.5731	0.0854
Fish	−0.0836	−0.3108	−0.2139	−0.1851	−0.1868	−0.1126	−0.2698	−0.2100	0.0485	−0.0576	−0.3594	−0.0147
Processed meats	−0.0660	0.3438	0.0620	0.1281	−0.0945	0.1996	0.4206	0.1589	−0.0258	0.3502	−0.0721	0.1580
Butter and margarine	0.0355	0.0189	0.6500	−0.0746	0.0523	0.0121	0.1632	0.5826	0.1076	0.6051	−0.0333	0.1782
Breads, cereals and crackers	−0.0095	0.0138	0.7400	−0.0124	−0.0820	−0.0322	0.0202	0.7324	0.0196	0.7421	−0.0356	−0.0279
Sweets and desserts	−0.1255	0.4280	0.1581	0.0845	−0.2056	0.1342	0.3564	−0.0479	−0.2163	0.1288	0.1328	0.4897
Snacks	−0.1761	0.4003	−0.3048	0.0792	−0.3064	0.2026	0.2000	−0.3542	−0.2242	−0.2037	0.0662	0.5412
Sugary drinks	0.0483	0.6404	−0.0807	−0.0791	−0.0316	0.0418	0.6446	0.0057	0.096	0.0551	−0.0066	0.7284
Dairy products	−0.0401	−0.1483	0.5637	0.1339	−0.0261	0.1754	−0.1603	0.5346	0.0199	0.4942	0.1519	−0.2213
Cheeses	−0.2155	0.1484	0.1091	0.5269	−0.1951	0.4756	0.2175	0.1766	−0.3495	0.2600	0.3486	−0.0729
Olive oil	0.0840	−0.1040	−0.0137	0.6043	0.0516	0.6379	0.1038	0.0009	−0.004	0.1101	0.5631	0.0240
Variance (%)	11.46%	10.63%	9.28%	8.05%	12.32%	9.91%	9.27%	7.71%	12.25%	9.76%	9.47%	8.14%

* Factor load ≥ 0.30 or ≤0.30.

**Table 4 nutrients-14-04716-t004:** Food patterns and factor loads of food groups extracted from the factor analysis with the principal component extraction method for individuals of the BALANCE program’s control group. Brazil.

Food Groups	Food Patterns—Baseline Year				Food Patterns—First Year of Follow-Up				Food Patterns—Second Year of Follow-Up			
	Traditional	Snack	Cardioprotective	Western	Traditional	Cardioprotective	Western	Snack	Traditional	Cardioprotective	Snack	Western
	Factor Load *											
Rice	0.8092	−0.0353	0.1162	−0.0107	0.7767	0.0023	−0.0950	−0.0262	0.7716	0.0429	−0.0505	0.0297
Pasta	−0.2046	0.0282	−0.1869	0.3245	−0.1580	−0.2167	0.3478	−0.0511	−0.1962	−0.0194	−0.0467	0.2836
Beans	0.7871	0.0058	−0.0509	0.0561	0.7139	−0.0242	−0.0059	−0.0391	0.6801	−0.0115	−0.0300	−0.036
Vegetables	−0.1025	0.1168	0.5026	−0.3194	−0.1376	0.5521	−0.1191	−0.028	−0.1377	0.5733	0.0615	−0.2435
Fruits and juices	0.1776	−0.0372	0.6898	−0.0274	0.2136	0.5625	0.1217	0.0816	0.2673	0.6202	0.0055	−0.0568
Meat and eggs	0.3919	−0.2389	0.0321	−0.0317	0.6020	0.0416	0.2101	0.0353	0.5985	0.0569	−0.0019	0.0964
Fish	0.0720	0.0880	0.0682	0.6170	−0.0427	−0.0873	0.4692	0.3557	0.1395	−0.2221	0.2431	0.3197
Processed meats	0.0201	0.0634	0.1355	−0.0970	−0.1483	0.1091	−0.0864	−0.0749	−0.1195	0.2141	−0.1711	−0.0057
Butter and margarine	0.1285	0.5144	0.0061	0.3229	0.1121	−0.0576	0.0168	0.5226	0.1571	−0.1075	0.3842	0.2802
Breads, cereals and crackers	−0.1105	0.6650	0.0951	0.1067	−0.0398	0.1074	−0.1102	0.7322	−0.0445	−0.0913	0.5289	0.1051
Sweets and desserts	−0.0273	−0.1202	0.0120	0.4320	−0.1639	0.0991	0.4203	0.0415	−0.093	0.0793	−0.2381	0.4181
Snacks	−0.1943	−0.1542	−0.0013	0.3006	−0.2218	0.0197	0.4950	−0.2359	−0.1138	0.0415	−0.2631	0.5664
Sugary drinks	0.0774	−0.0930	−0.1972	0.4945	0.1123	−0.0252	0.7123	−0.0732	0.1246	−0.0205	0.1117	0.7166
Dairy products	−0.0343	0.7033	−0.0672	−0.1694	−0.1151	0.5571	−0.2174	0.0641	−0.1406	0.1762	0.7161	−0.0482
Cheeses	−0.1827	0.2461	0.3945	0.2735	−0.2232	0.3950	0.2884	0.3782	−0.1935	0.5815	0.1067	0.2725
Coffee and tea	−0.1034	−0.3592	0.1813	0.2296	−0.0884	−0.4414	0.0108	0.4326	−0.0747	−0.0496	−0.5422	0.0677
Olive oil	0.0196	−0.0603	0.6099	0.0820	0.2083	0.4024	0.1781	0.1634	0.2418	0.5639	0.0884	0.0478
Variance (%)	11.67%	10%	9.04%	7.95%	12.02%	10.34%	9.10%	8.13%	11.83%	9.88%	9.18%	7.80%

* Factor load ≥ 0.30 or ≤0.30.

**Table 5 nutrients-14-04716-t005:** Food patterns of the intervention group according to socioeconomic, nutritional status and lifestyle characteristics in the base year and in the second year of follow-up of the BALANCE program. Brazil.

Variables	Baseline Year				Second Year of Follow-Up			
	Western	*p*	Cardioprotective	*p*	Western	*p*	Cardioprotective	*p*
Sex								
Male	0.113 (1.06)	<0.001 *	0.074 (1.03)	0.002 *	0.121 (1.03)	<0.001 *	0.092 (1.02)	0.001 *
Female	−0.166 (0.86)		−0.106 (0.93)		−0.179 (0.91)		−0.139 (0.94)	
Age								
Adult	0.093 (1.05)	<0.001 *	0.025 (1.01)	0.312	0.056 (1.06)	0.067	0.045 (0.98)	0.142
Elderly	−0.126 (0.90)		−0.034 (0.98)		−0.074 (0.90)		−0.059 (1.01)	
Economic Class								
A/B	0.031 (1.01)	0.215	−0.049 (0.95)	0.600	−0.008 (0.94)	0.361	−0.121 (0.95)	0.057
C/D/E	−0.055 (1.02)		−0.084 (0.94)		−0.080 (0.95)		0.030 (0.97)	
BMI								
Underweight	0.129 (0.64)	0.539	−0.406 (0.70)	0.037 *	0.073 (0.99)	0.154	−0.181 (0.79)	0.099
Eutrophic	−0.050 (0.97)		−0.069 (1.00)		−0.134 (0.83)		0.164 (0.96)	
Overweight /obesity	0.001 (1.01)		0.035 (1.00) ^a^		0.040 (1.01)		−0.014 (0.95)	
WC								
Adequate	0.234 (0.82)	0.071	−0.308 (0.64)	0.020 *	0.221 (1.08)	0.202	0.200 (1.10)	0.248
Inadequate	−0.016 (0.99)		0.019 (1.01)		−0.009 (0.99)		−0.008 (0.99)	
FA								
Active	0.044 (1.01)	0.254	−0.045 (0.95)	0.924	−0.042 (0.94)	0.153	0.049 (0.93)	0.269
Inactive	−0.029 (0.99)		−0.039 (0.96)		0.075 (1.01)		−0.037 (1.00)	
Smoking								
Never smoked	−0.056 (0.94) ^a^	0.011*	−0.019 (0.96)	0.853	−0.077 (1.00)	0.178	0.077 (1.01)	0.140
Former smoker	0.001 (1.01)		0.013 (1.05)		0.053 (0.99)		−0.065 (1.00)	
Smoker	0.295 (1.17)		0.016 (0.81)		0.083 (1.01)		0.059 (0.85)	

^a^ different groups according to the Bonferroni test. * *p*-values significant according to the Student *t*-test and ANOVA.

**Table 6 nutrients-14-04716-t006:** Food patterns of the control group according to socioeconomic, nutritional status and lifestyle characteristics in the base year and in the second year of follow-up of the BALANCE program. Brazil.

Variables	Baseline Year				Second Year of Follow-Up			
	Western	*p*	Cardioprotective	*p*	Western	*p*	Cardioprotective	*p*
Sex								
Male	0.158 (1.08)	<0.001 *	0.049 (1.04)	0.048	0.144 (1.08)	<0.001 *	0.032 (1.07)	0.268
Female	−0.212 (0.83)		−0.067 (0.92)		−0.210 (0.828)		−0.047 (0.87)	
Age								
Adult	0.078 (1.00)	0.002 *	0.003 (0.99)	0.912	0.063 (1.02)	0.031 *	−0.030 (0.98)	0.309
Elderly	−0.107 (0.98)		−0.004 (1.01)		−0.093 (0.95)		0.044 (1.02)	
Economic Class								
A/B	−0.226 (0.98)	0.414	−0.030 (0.99)	0.120	−0.010 (1.01)	0.704	−0.045 (0.96)	0.500
C/D/E	0.034 (1.010		0.078 (1.02)		−0.042 (0.90)		0.008 (0.94)	
BMI								
Underweight	−0.122 (0.95)	0.132	0.073 (1.01)	0.326	−0.268 (0.97)	0.236	0.100 (0.90)	0.763
Eutrophic	−0.090 (1.00)		−0.072 (0.91)		0.128 (1.04)		−0.027 (0.92)	
Overweight /obesity	0.037 (0.99)		0.025 (1.02)		−0.021 (0.98)		0.031 (1.02)	
WC								
Adequate	−0.071 (0.88)	0.566	−0.001 (0.95)	0.986	0.267 (1.13)	0.182	−0.211 (1.02)	0.159
Inadequate	0.005 (1.01)		0.002 (1.00)		0.031 (0.99)		0.031 (0.99)	
FA								
Active	0.103 (1.07)	0.009*	−0.028 (0.96)	0.647	0.015 (0.98)	0.281	0.041 (0.98)	0.316
Inactive	−0.062 (0.95)		0.001 (1.02)		0.110 (1.05)		−0.044 (0.96)	
Smoking								
Never smoked	−0.106 (0.90) ^a^	<0.001 *	0.006 (0.99)	0.994	−0.054 (0.98)	0.012 *	0.099 (1.03)	0.090
Former smoker	0.022 (1.01)		0.002 (0.98)		−0.010 (0.96)		−0.048 (0.98)	
Smoker	0.332 (1.24)		−0.005 (1.13)		0.385 (1.29)		−0.135 (0.89)	

^a^ different groups according to the Bonferroni test. * *p*-values significant according to the Student *t*-test and ANOVA.

## Data Availability

The data presented in this study are available on request from the corresponding author. The data are not publicly available due to privacy restrictions.

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
