# Peer review of "Tracking of Dietary Patterns in the Secondary Prevention of Cardiovascular Disease after a Nutritional Intervention Program—A Randomized Clinical Trial"

_nutrients, 2022, doi:10.3390/nu14224716_

Round 1

Reviewer 1 Report

-There are multiple grammar errors. A native speaker with expertise in the field should proofread the manuscript.

-Abstract: after line 17, add the content of the intervention in brief.

                  Line 26, no statistical analysis was done to compare the cardioprotective dietary pattern variances between the intervention group and control group. It “looks” increase in the intervention group than the control group, but the authors did not perform the statistical analysis.

-lines 220-221, double-check the # of subjects in the first year and second year follow-ups. The numbers are not shown/gotten in figure 1.

-Tables 1 & 2 should be switched.

-Table 2, HDL cholesterol standards are different from gender. Therefore, the data should be displayed by gender. Same for waisit circumference.

-Line 328-329, there was no significant change in age in the cardioprotective pattern at both baseline and the second year follow-up.

-Discussion: In Table 3, the variance observed for “snack” pattern in the intervention group after the first year was 7.71%. However, it went UP to 9.76% after the 2nd year. It should be discussed why the variance observed for snack pattern, positive loads for foods with low fiber and high fat contents, increased even after the intervention.

-Statistics: post-hoc tests should be conducted for ANOVA tests. Also, the group differences after the post-hoc tests should be displayed in tables 5 & 6.

Author Response

Dear Editor and Reviewers, 

Thank you for giving us the opportunity to review our article titled “Tracking of Dietary Patterns in the Secondary Prevention of Cardiovascular Disease after a Nutritional Intervention Program – a Randomized Clinical Trial.” We thank you and the reviewers for their valuable insights and suggestions, which we believe have strengthened our work.  Below we describe our answers to your concerns and indicate where in the manuscript we made the changes.

Looking forward for your decision. 

Best regards.

Reviewer 2 Report

The manuscript comparatively analyses the efficacy of different dietary patterns to improve the eating habit and secondary cardiovascular prevention in the Brazilian population.  Studies aimed to know better response of the population to programs addressed to improved cardiovascular health are of high relevance at the general population and clinical level. The present study refers to 2 different arms with different dietary assessment and 4 types o dietary patterns in each arm.   

- A Major concern refers to the different food-pattern distribution at baseline between the control and intervention groups. Authors need to analyze and discuss how this might affect their findings. In this respect last paragraph in page 8 (lines 301-307) needs to be thoroughly revised. The only common aspect is the “traditional” pattern, which is the most frequent in the Brazilian study population regardless the study arm and year of follow-up.   Accordingly, the main objective of the study should be rewritten, authors should not compare evolution of dietary patterns between control and intervention group due to the different distribution at baseline.  

- Is there any effect of the intervention program on relation to the adherence to the food groups within the different food patterns compared to the control group?

- To facilitate reading of the manuscript authors should indicate in text which results in tables support results given in lines 342-354.

- According to the authors, the study refers to secondary prevention. Please indicate the number (or %) of subjects with each of the cardiovascular pathologies indicated in page 4-5.

- Compared to baseline, were there differences in the evolution of the variables indicated in table 2 between the control and intervention groups at 2 years of follow-up?  This needs to be considered in the manuscript.

- How can authors explain the high increase in cholesterol, LDLc , triglycerides and glycemia in the study population after 2 years intervention? This needs to be carefully considered and discussed

- Authors need to differentiate in the discussion section, which results are shown in the present manuscript and which derive from the previous work (e.j. ref 21)

- Main conclusion needs to be revised since control and intervention groups had a different variance for the cardioprotective eating pattern at baseline.

- Discussion should be shortened. Some information is already given in previous sections. To facilitate the reading avoid providing new information in the discussion section and focus it to discuss findings provided in the discussion section.

Minor comments

- The terms no risk for CVD and “with risk for CVD” in table 2 should be revised since all subjects were in secondary prevention.

- To facilitate reading, authors should briefly sumarize the inclusion / exclusion criteria of the study, protocol and randomization procedure (ref 19, 21, 22).

- Text in page 3, lines 130-134 should be move to the results section

- Expression “intervention group” seems more appropriate than “experimental group” . Please change it in the manuscript

Author Response

Dear Editor and Reviewers, 

Thank you for giving us the opportunity to review our article titled “Tracking of Dietary Patterns in the Secondary Prevention of Cardiovascular Disease after a Nutritional Intervention Program – a Randomized Clinical Trial.” We thank you and the reviewers for their valuable insights and suggestions, which we believe have strengthened our work.  Below we describe our answers to your concerns and indicate where in the manuscript we made the changes.

Looking forward for your decision. 

Best regards,

Round 2

Reviewer 1 Report

The manuscript has been sufficiently improved